# Uncertainty and precaution in hunting wolves twice in a year

**Adrian Treves** **\*, Naomi X. Louchouarn**

Nelson Institute for Environmental Studies, University of Wisconsin, Madison, Wisconsin, United States of America

\* atreves@wisc.edu

**Data Availability Statement:** : All relevant data are within the paper and its Supporting Information files.

**Funding:** AT was funded by DAAD: German Academic Exchange Service (Personal reference no.91803225). NXL was funded by the Natural

## Abstract

When humanity confronts the risk of extinction of species, many people invoke precautions, especially in the face of uncertainty. Although precautionary approaches are value judgments, the optimal design and effect of precautions or lack thereof are scientific questions. We investigated Wisconsin gray wolves *Canis lupus* facing a second wolf-hunt in November 2021 and use three legal thresholds as the societal value judgments about precautions: (1) the 1999 population goal, 350 wolves, (2) the threshold for statutory listing under the state threatened and endangered species act, 250 wolves; and (3) state extirpation <2 wolves. This allows us to explore the quantitative relationship between precaution and uncertainty. Working from estimates of the size wolf population in April 2021 and reproduction to November, we constructed a simple linear model with uninformative priors for the period April 2021-April 2022 including an uncertain wolf-hunt in November 2021. Our first result is that the state government under-counted wolf deaths in the year preceding both wolf-hunts. We recommend better scientific analysis be used when setting wolf-hunt quotas. We find official recommendations for a quota for the November 2021 wolf-hunt risk undesirable outcomes. Even a quota of zero has a 13% chance of crossing threshold 1. Therefore, a zero death toll would be precautionary. Proponents for high quotas bear the burden of proof that their estimates are accurate, precise, and reproducible. We discuss why our approach is transferable to non-wolves. We show how scientists have the tools and concepts for quantifying and explaining the probabilities of crossing thresholds set by laws or other social norms. We recommend that scientists grapple with data gaps by explaining what the uncertainty means for policy and the public including the consequences of being wrong.

## Introduction

When humanity confronts threats to the planetary or local natural resources and biodiversity, many governments, critics, and commentators invoke precautions. For example, in 1992, United Nations authors endorsed a precautionary principle as follows,

> "In order to protect the environment, the precautionary approach shall be widely applied by States according to their capabilities. Where there are threats of serious or irreversible

Sciences and Engineering Research Council of Canada (PGSD3 - 545968 - 2020). Neither source of funding was for this work specifically. This does not alter our adherence to PLOS ONE policies on sharing data and materials.

**Competing interests:** The authors declare no financial competing interests. AT discloses the following non-financial, potential competing interests. Professional service to organizations or editorial boards Board of director (unpaid): President, Future Wildlife (2020), Board member Wildlife for All (Sep. 2021-present) Science advisor (unpaid): Project Coyote (2012–) Northeast Wolf Coalition (2014–) Endangered Species Coalition (2016–) Friends of the Wisconsin Wolf (2015–) Living with Wolves (2016–) Rocky Mountain Wolf Coalition (2018–2021) Earth and Animal Advocates (2019–) Benton County's Agriculture and Wildlife Protection Program (2018–) Wild Earth Guardians (2020–) Member (unpaid): Union of Concerned Scientists (2015–), IUCN Bear Specialist Group task-force on human-bear conflicts (2012), IUCN Wolf specialist (2016–), Public Employees for Environmental Responsibility (2015–2019). Expert declarations (unpaid): Wi Federated Humane Societies et al. v Stepp. 2013. WI Court of Appeals District IV; WEG v Colorado Parks and Wildlife Commission et al. 2017. District Court, Denver Country, Colorado; Western Watersheds Project et al. v USDA Wildlife Services. 2018. U.S. District Court for the District of Idaho 1:17-cv-00206-BLW Doc 22-3; CBD & Cascadia Wildlands v WDFW 2018. Superior Court of Washington for Thurston County. 18-2-04130-34. CBD v WDFW et al. 2019. Superior Court of Washington for Thurston County, 18-2-02766-34. Huskin et al. v WDFW et al. 2019. Superior Court of Washington for King County 19-2-20227-1 SEA. Great Lakes Wildlife Alliance et al. v. Cole et al. Circuit Court Dane County, Wi 2021cv002103 Branch 9 Circuit Court Summons Dane County, WI, Case 2021CV002103 Document 5 Filed 08-31-2021 Paid service as external reviewer or speaker: Swiss-NSF SPARK (2019), Antioch University of New England (2018–2020), Landmark Foundation (2017), various publishers (2007–2017), U.S. Fish & Wildlife Service (2019), French Ministry of Environment, Scientific Council on Wolves (208-present), Ministry of Environment, Alfred Toepfer Academy for Nature Conservation, Lower Saxony, Germany (2021-present), NABU, Germany (2015, 2021) This does not alter our adherence to PLOS ONE policies on sharing data and materials.

damage, lack of full scientific certainty shall not be used as a reason for postponing cost-effective measures to prevent environmental degradation." (Principle 15 of [1]).

## Precaution

The precautionary principle can be a double-edged sword. For many fields harm can arise from action or inaction, so the task of implementing precautions is not always obvious. For many practitioners debating whether to intervene in human poverty or illness, inaction can kill. Therefore, the harm and the precaution are not necessarily obvious. (For a full treatment of the precautionary principle or approach in fields from civil engineering to medicine, we recommend this article [2]). Where poverty or illness are the major killers, technological and medical interventions that alleviate these ills can save lives, and therefore, inaction can perpetuate harm. The precautionary principle seems to us more straightforward to apply when the potential harm is extinction.

There is no scientific uncertainty that human activities that directly kill organisms or degrade ecosystems have caused extinctions. The risk of extinction whether local or range-wide is higher for organisms that are few in number, or abundant ones that are narrowly endemic or genetically homogeneous [3]. For simplicity, we refer to the latter as listed hereafter. Precautions for imperiled species received affirmation by the 1978 USA Supreme Court decision on the snail darter threatened by Tellico Dam [4]: "The Supreme Court's opinion in TVA v Hill is still good law, with Chief Justice Burger's stentorian declaration repeatedly echoed in successive endangered species cases: 'Congress has spoken in the plainest of words, making it abundantly clear that the balance has been struck in favor of affording endangered species the highest of priorities, thereby adopting a policy which it described as **Institutionalized caution**.'" p.305, emphasis added [5], citing majority opinion [4]; see also [6]. For example, under Endangered Species Act (ESA) protections and similar provisions of the E.U. Habitats Directive [7–9], permits for killing listed species are extremely restrictive.

Following efforts to reduce protections for gray wolves *Canis lupus* in the USA and E.U., much attention has been paid to proposed and enacted regulations and methods for public hunting, trapping, and hounding of wolves [10–20]. For wolves in the USA, a recently listed population reclassified from ESA endangered status in early January 2021, but whose reclassification is a matter of litigation as we write [21], similar institutionalized caution might still be appropriate. For example, in the wake of USA federal de-listing, the state of Wisconsin held a wolf-hunt in February 2021 during which permitted hunters killed at least 21% of the population in <72 hours [22]; another 98–105 wolves were estimated to have died (from poaching mainly) because of removal of federal protections between 3 November 2020–14 April 2021; and apparently at least a third of collared wolves went off the air without explanation [23,24]. A March 2021 proposal to hunt Wisconsin wolves again starting 6 November 2021 has raised public concerns and state wildlife agency cautions to decision-makers [25].

Here we present the second in a series examining the effects of wolf-hunting on Wisconsin's wolf population [23] by forecasting the status of the population out to 14 April 2022, with and without permitted killing at various levels. To operationalize precaution without interposing our own values, we defined the result of wolf-hunting by the state of Wisconsin as eradication (<2 wolves), statutory listing under the state threatened and endangered species list (<251 wolves), and falling below the state population goal of 350 wolves [26]; all those values exclude wolves ranging across tribal reservations estimated at 42 wolves [27]. These three thresholds represent the value judgments made by society at one time or another, in principle, statute, and regulation respectively, about how cautious one should be about the status of the

state wolf population. We are not interposing our own value judgment about a desirable or undesirable number of wolves. Instead, we ask the scientific question of what death toll in Fall 2021 would cross undesirable thresholds set by existing regulatory mechanisms, so the public and decision-makers can judge caution and its absence.

Scrutiny of this case allows both a qualitative and a quantitative analysis of uncertainty in the presence or absence of institutionalized caution. Our interest in scrutinizing these plans is not ours alone. The federal legal mandate is 5 years of monitoring and possible emergency relisting under the ESA if the threats to wolves resurface strongly [28]. Given that the state wildlife agency expects serious federal scrutiny if the state population is reduced by 25% and recommended a lower quota of wolf-kills preceding both wolf-hunts than was set by the Natural Resource Board, NRB [25] and given co-sovereign tribes in the region have expressed strong concerns [29], scrutiny of the plans for a second wolf-hunt seems important to many actors. Relatedly, concerns have been expressed by scientists and managers about 'political populations' defined as wildlife whose population parameters are set by political pressures despite being biologically unrealistic [30]. Scientific work that can bridge between biological (or social scientific) observations on the one hand, and management or policy-making on the other hand, may help to minimize undue political pressure. Scientific scrutiny also presents a case study of the precautionary principle in the design of sustainable natural resource use.

## Uncertainty

The U.N. precautionary principle 15 above calls for reducing scientific uncertainty. Likewise, an early amendment to the USA ESA sought to base decisions solely on "the best available scientific and commercial data", BAS [5]. Those principles identify scientific certainty and uncertainty as crucial fulcrums for decisions with more deliberation and less action the more uncertain we are.

When precautionary approaches are reduced to a question of certainty about harms, policy-makers face a dilemma well summed up in this quotation, "The very basis of the Precautionary Principle is to imagine the worst **without supporting evidence** . . . those with the darkest imaginations become the most influential." emphasis added, [31]. To avoid that pitfall which afflicts extreme positions in the wolf-hunting debate, we do not imagine the darkest future but rather stick to peer-reviewed data and, where that is absent, restrict ourselves to the official state data, rely on peer-reviewed evidence when it conflicts with the state's assertions of fact, and explain the limits to confidence with both.

The uncertainties in our case are not limited to scientific data or how to interpret those. The uncertainties extend to the political actors and decision-makers. Powerful actors differ on the ideal number of wolves dead or alive and competing views of what makes for the best available science. The socio-political context of the Wisconsin wolf debate includes multiple governmental entities, each one with a different worldview and each one able to act (subsequent to our writing) in ways we cannot anticipate. Given these actors differ in their institutionalized caution and in how individuals are given authority to use personal opinion about caution, our three above-mentioned thresholds (eradication, listing level, and population goal) serve as legal value judgments about precautions. Hence, the legal thresholds provide the basis we use to account for uncertainty.

Uncertainty also characterizes the scientific literature on human-induced mortality patterns among wolves. We do not spend much effort to address sustainability for two simple reasons. First, concerns with sustainability are about future uses more than the risk of extirpation after a single use and we are concerned with crossing the above thresholds in the 2021–2022 wolf-hunting season. Second, the science of sustainable hunting of wolves is unsettled. Although

reviews of wolf population dynamics and sustainable levels of killing include many data points and seem to converge on a range of sustainable, annual human-caused mortality rates [32–36], the literature nonetheless concludes with three-fold differences in magnitude for estimates ranging from high teens to 48%. Although the prior literature would seem to guide decision-makers in Wisconsin to choose a Fall 2021 wolfs-hunt quota that would not change the population, the wide variation in estimates above and the novelty of a second wolf-hunt in a single year produces new and greater uncertainties than the literature addresses. Also, in a series of papers on wolf science and policy in Wisconsin, we have shown how omissions of a history of methodological changes in censuses, censoring the information available in the disappearances of marked wolves, and a lack of alternative management scenarios altogether could both distort wolf policy and mire the science in uncertainties due to methods [23,33,37–46].

To support decision-making in the face of great uncertainty, we provide a step-by-step rationale for the uniform distributions we use and a simple linear model of births and deaths. The primary reason to take this simple approach is its practical advantage. We show how the state, tribes, public, and other interests can perform these estimates independently and reproduce our findings to explore their own scenarios for November death tolls. That is valuable given our inability to predict the eventual death toll and the reactions of the many interested governmental actors mentioned above. Thus, as we grapple with uncertainty at every step, we transparently present the bounds we consider plausible and why. Secondly, we use Bayesian concepts and terminology but not formal Bayesian algorithms, because many of our key input variables are uninformative and combine in simple linear fashion. To achieve our primary goal of clear communication and user-input, a formal Bayesian algorithm would be less accessible. We illustrate how any reader and user of our simple model can choose a death toll and calculate probabilities of crossing the legal thresholds. We offer this simple approach as a possible model for other scientists engaged in public policy debates whether or not contentious and uncertain, beyond wolves, and beyond North American hunting systems.

## Materials & methods

Our study period is the wolf-year starting 15 April 2021 and ending 14 April 2022. We contended with three key scientific uncertainties in this study period. First, the effects of the 22–24 February wolf-hunt on wolf numbers, pack sizes, and reproductive potential are uncertain. Second, little information is available about reproduction for our study period. Wolf reproduction data is generally difficult to collect and the state census method used tends to confound pack size with past reproduction [46,47]. Third, we could not be confident about the legal quota when we analyzed data in Fall 2021 nor does anyone know the eventual death toll. Therefore, our forecasts for 14 April 2022 include estimates of all wolf mortalities even if the legal quota ends up unfilled. We describe the unprecedented methods of the February 2021 wolf-hunt first because it conditions the remaining uncertainties.

The February 2021 wolf hunt killed 218 wolves legally, took place during the mating and pregnancy season of the wolves, and included pursuit in deep snow by snowmobiles, night-time hunting, hounds in packs of 6, and relays that allowed a team of hunters to substitute a fresh pack of hounds; >85% of kills were aided by the use of hounds according to hunter self-report [22]. Hunters overshot the legal quota by 99 wolves (82%), an event the DNR blamed on regulations that require 24 h notice to close zones and regulations that allowed hunters in open zones to delay reporting kills for 24 h even after the state quota was met. Also, the state sold permits for 13 hunters for every wolf that could be legally killed. These latter regulations increase the uncertainty about the eventual death toll of any legal quota [22,23,25,48].

Before we address the remaining uncertainties about population status in our study period, with a mix of qualitative and quantitative information, we explain the simple model we adopted for population change during the study period. Because of the preceding three scientific uncertainties and our desire to provide a method that others can use to plug in their own values or future data, we relied on a simple one-step model of population size change for our study period, as follows:

$$N_{t+1} = N_t + R_t - M_t - H \qquad (1)$$

where $N_t$ is the population size estimate on 15 April of year t, t = 2021, Rt is the number of pups born in year t surviving to November when they are typically counted alongside adults using standard census methods [35,49], H is the death toll in a wolf-hunt, and $M_t$ is the number of dead wolves in year t. We estimated $R_t$ by Eq 2,

$$R_t = B_t * L * S \qquad (2)$$

where $B_t$ is the number of breeding packs, L is the litter size, and S is pup survival. We estimated $M_t$ by Eq 3,

$$M_t = D \bullet (N_t + R_t / 2) \qquad (3)$$

where D is the annual mortality rate estimate for a year without ESA protections and without a wolf-hunt as we describe further below in the section on deaths. Note that R from Eqs 1–3 represents pups surviving to November 2021. In Eq 3 these pups are exposed to one-half of a year of D from November-April.

Our simple model in Eq 1 assumes no net migration into or out of the state during the study period at a rate relative to deaths or births substantial enough to affect our results. Assuming no net migration is a precaution because it would be hopeful to imagine rescue from outside the state if legal thresholds were crossed in the state. Our assumption seems reasonable given long-distance migration leading to pack establishment has been rare [50]. Also, the assumption of no net migration has been used by others modeling this population [51,52]. Also, Eqs 1–3 assume linear effects. We assumed no compensatory increases in birth or pup survival other than those encompassed by the range of values in [53]. We do not ignore Allee effects, compensation or negative density-dependence [54,58,59], but we do not model them because too many questions remain for Wisconsin wolves [3,41,43]. Nor do we model non-linear effects that would caution against high death tolls in a second wolf-hunt. For example, depensatory or super-additive effects as described by numerous studies of wolves including in the Wisconsin wolf population [33,36,45,60,61]. We defend the simplicity of our approach as follows: pending evidence that non-linear effects would play out detectably in the short period of our study and pending an analysis of net compensatory and depensatory effects, we simply assume the good conditions studied by [56] encompass any nonlinear effects for wolves in an environment with fewer competitors than before.

## Population size estimation

The second source of uncertainty described above was the point estimate and precision of that estimate of population size. The state government had implemented a new, unpublished method of census (hereafter new census method) which produces systematically higher estimates than the traditional census method [27,54,55]. However, the unprecedented February hunt described above, interrupted that census. Ending wolf census on 21 February has never been done. The resulting uncertainty about $N_{2021}$ leaves us with two estimates using two methods.

The state estimated $N_{2020}$ by two methods, following [27]. The old census method yielded 1034–1057 (uninformative uniform distribution). Used since 1979 with a few changes over time, the traditional method attempted complete enumeration referred to as a minimum count [56], although efforts to validate that it did not double-count wolves are still lacking. The second, new census method yielded 1195 (957–1573, unknown distribution) and used an occupancy framework but the method has still not been published in a peer-reviewed, transparent manner [24]; S1 Fig 1. Although the two methods differ substantially in uncertainty, they don't result in very different point estimates for $N_{2021}$.

The state and [23] estimated $N_{2021}$ in two ways. We estimated it from the old census method and estimates of population growth parameters and estimates of annual mortality rates [23] at 695–751 wolves, which we considered a maximum because of the likelihood of greater rates of illegal killing given the conditions of that hunt summarized above. The second estimate of $N_{2021}$ comes from the state government in summer 2021 and uses the new census method interrupted at 21 February 2021 [25].

The state's justification for interrupting the new census method before 14 April 2021, when it would have been terminated as in previous years [27], was that the wolf-hunt of 22–24 February made accurate and precise data collection impossible. Therefore, the wolf population estimate derived from the new census method in 2021 lacked non-hunt mortality from 25 February to 14 April 2021, which is a season of high mortality from winter conditions and illegal killing historically [39,57–59]. We are not aware of any effort to correct the new census method estimate, therefore it seems to be a systematic over-estimate of $N_{2021}$. Furthermore, the state did not provide bounds on $N_{2021}$ but given the reported value (1195) of $N_{2021}$ equaled the central tendency of $N_{2020}$ (also 1195), we assume here the same bounds minus the 218 wolves killed legally in the February wolf-hunt, hence 977 (739–1355). That value minus some unaccounted late winter mortality would bring the estimate closer to the prior estimate of 695–751. But the similarity of the two estimates for $N_{2021}$ is hard to evaluate so we use both throughout.

## Reproduction

Eq 1 required the number of pups surviving to November, which in turn, requires Eq 2 to produce an estimate of B for the number of breeding packs, L for litter size in mid-summer, and S, pup survival to November. Because we face a nearly complete absence of information on wolf pack reproduction in summer 2021 [25,48], we used a mix of informative priors for L, S, and the proportion of potentially reproductive pairs that actually bred.

We used the only peer-reviewed, published study of reproductive success before November conducted among Wisconsin wolves [53], which provided estimates for the proportion of packs producing litters (0.55–0.89, mean 0.72), for L, litter size (3–6, mean 4.8), and for S, pup survival to 3–9 months 0.05–0.72 with a mean of 0.2, from three separate normal distributions centered on the means and bounded by the 95% CI around those means. For pup survival to 3–9 months, we noted the long right tail of the distribution in [53] and adjusted the normal distribution accordingly. Hence multiplying the three preceding parameters yielded an average of 0.69 (95% CI 0.15–4.32) pups surviving to Novemebr per pack. We estimate the number of breeding packs, B, to multiply it against in the following section.

The study in [53] was conducted during a period with ESA protections and a population recolonizing vacant range, i.e., reproductive performance in good years measured by [53]. We did not use another commonly cited summary [56] because it aggregated breeding data at the end of the wolf-year in April and we needed an estimate for November. Also, we have previously explained why winter estimates of pack size might be confounded with estimates of breeding at that time [47].

Number of breeding packs, B: The proportion of packs that produced pups in summer was estimated in [53] as a proportion of all packs studied. We had to estimate B from the packs present in the state multiplied by Thiel's [53] estimate of the proportion producing a litter. For summer 2021, we assumed that the former was some subset of the total number of breeding females surviving the February 2021 wolf-hunt. For summer 2020, we used [53] estimates and a highly informative prior as follows.

In April 2020, the state contained 245 packs and tribal reservations held 11 packs [27]. An unknown number were eliminated in the February 2021 wolf-hunt. The state assumed no disruption to breeding after the February 2021 wolf-hunt [25]. Given the unprecedented nature of the wolf-hunt, the effects of the February 2021 wolf-hunt on R are uncertain. The number of packs that produced pups in summer 2021 might have been strongly affected by the February 2021 wolf hunt that took place during the breeding season and used methods (hounds, snowmobiles, night-time tracking) that might have made breeders more vulnerable than in prior wolf hunts. Given the urine-marking habits of territorial alphas in snow, the possible olfactory conspicuousness of reproductively active alphas in February, the use of hounds, some but not all of our scenarios below treat breeding females as relatively more vulnerable than pack-mates and more vulnerable than in past years.

Reproductive success of wolf packs might drop when humans kill pack members, either directly through death of breeders or indirectly through stress, loss of adult wolf helpers, wounding, or other factors caused by people. Although there is high variability in the effect of breeder loss across studies and time of year [60–63], it is clear that breeders killed during the pregnancy or mating season almost invariably result in reproductive failure of the entire pack, especially when the alpha female dies. There is less evidence for the effect of removing other wolves, the effect of the novel methods used in the February 2021 wolf-hunt, or the effect of poaching on subsequent reproductive success of wolf packs. These data are almost absent for Wisconsin (but see [61]). Therefore, we estimated the number of breeding packs (B) in several ways.

We have five sources of information that help to parametrize B the variable of number of breeding packs in summer 2021. First, under beneficent conditions studied by [53], we know the mean (95% CI) for the proportion of packs that bred was 0.72 (0.55–0.89) during early to middle colonization under ESA protections during a less politically contentious phase of wolf policy. It seems inconceivable that a greater proportion of packs could have bred in summer 2021, so 218.05 (0.89 x 245 packs across the state) seems like an appropriate starting point to deduct packs that failed to breed because of the February 2021 wolf-hunt.

The minimum plausible deduction from 218.05 is 51 breeding packs which corresponds to approximately 0.23 pregnant females per wolf-kill. Below we explain why this is a minimum plausible deduction from 218.05. A preliminary report from a sample of 22 wolf carcasses volunteered by hunters from the February 2021 wolf-hunt was necropsied by the Great Lakes Indian Fish & Wildlife Commission [64]. They reported 65% of adult females and 50% of yearling females were pregnant in that small, nonrandom sample. Our minimum plausible proportion of 23% is much lower because a larger sample from a different hunt in Fall 2012 in neighboring Minnesota suggested 0.20–0.25 wolves were females with evidence of past breeding [65]. This hunt was very different (no hounds, no deep snow, no snowmobiles, no nighttime hunting, not during mating season, etc.). Given the average pack size in our region in late winter is approximately 4 wolves with a longer right tail (2–12), it would appear somewhat less than a quarter of pack members would be pregnant females if hunters killed them in proportion to their presence in the population. Thus deducting 51 wolf packs is one-quarter to one-sixth of the 218–323 extra deaths we described above. That leaves B = 167 as the maximum plausible upper bound.

The maximum plausible value of B described above seems a maximum for several reasons. For one, the Timber Wolf Alliance and Timber Wolf Information Network conducted summer

2021 howling surveys in portions of the state and estimated that fewer than half of the packs they encountered responded with pup vocalizations [64] citing court declaration by A.P. Wydeven. Such howling surveys are somewhat accurate for the detection of pups in experimental, field tests but are not accurate for counting pack size or pup numbers in those same tests [66]. Although we cannot extrapolate to the whole state or assume that response to human howls would continue as in the past, their anecdotal data suggest a scenario with a lower estimate is also plausible. Also, there are reasons to expect breeding females would have been selected in greater proportions than their representation. Pregnant or mating female wolves deposit blood and different hormonal odors in their urine left to mark territorial boundaries. The large number of hounds used in the February 2021 wolf-hunt with deep snow might have made breeding females particularly conspicuous. Then we might use the higher value from Red Cliff instead to estimate that 144 wolf packs failed to reproduce in summer 2021, leaving B = 74 as a plausible lower bound. However, we suspect the real value lies between B = 74–167.

We also used an indirect source of information which came from spatial analysis of kill locations in February 2021 wolf-hunt to generate two additional scenarios. We assume that wolf packs that might have encountered hunters or hounds during the February 2021 wolf-hunt might be disrupted reproductively by stress or deaths of pack-mates. We assumed the maps of hunted areas and pack areas were accurate, every pack near to a hunted area would potentially be affected by hunting, and reservation packs and packs outside of hunted counties would be unaffected by hunting. If the spatial proximity of reported wolf-kills predicts the disruption of reproduction in the nearest pack, then the two scenarios in Fig 1 provide two more estimates of the number of breeding packs.

Note our unlikely lower bound of 12 breeding packs emerged from scrutiny of Fig 1 because only one pack lay mainly in a county without reported kills and 11 other packs lay mainly in tribal reservations where hunting was prohibited [64]. If hunters exert a suppressive effect on reproduction of wolf packs in a large area, the number of breeding packs would be estimated by B = 91. That is equivalent to 0.41 of our unlikely upper bound or the failure of 127 packs to breed. If hunters exert a suppressive effect in a much smaller area, the number of breeding packs would be estimated by B = 129.

In sum, we found four point estimates of the number of breeding packs that seem plausible (74, 91, 129, 167) without any additional information to choose between them. In Fig 2, we represent the uninformative uniform distribution between those four values and implausible, extreme values of 12 and 218.05.

## Deaths

Eq 1 requires an estimate of $M_{2021}$, the number of dead wolves (composed of adults year-round and pups after November 2021), which relied on an estimate and variation in the annual mortality rate (D) as an input to Eq 3. We began by solving Eq 1 for M and R in year t = 2020. Because we knew N for t and t+1, Eq 1 reduces to a change in population equals births minus deaths. Also, we had an informative prior $R_{2020}$ from [53] for a summer with ESA protections following a winter with no wolf-hunt. Hence, we solved for $M_{2020}$, which we used as an input to Eq 2 for D, the range of annual wolf mortality rates for years with those conditions. Note we did not use multiple prior years to estimate D because the last 5 years were under strict ESA protections year-round unlike 2020–2021, nor did we use the years with wolf-hunts 2012–2014 because these lacked one or both of the conditions in February 2021 (hunting with hounds or deep snow cover during the wolf mating season).

We present the estimates of D in Results but validating these may not be obvious. There is little scientific consensus on annual mortality rates among Wisconsin wolves. The DNR

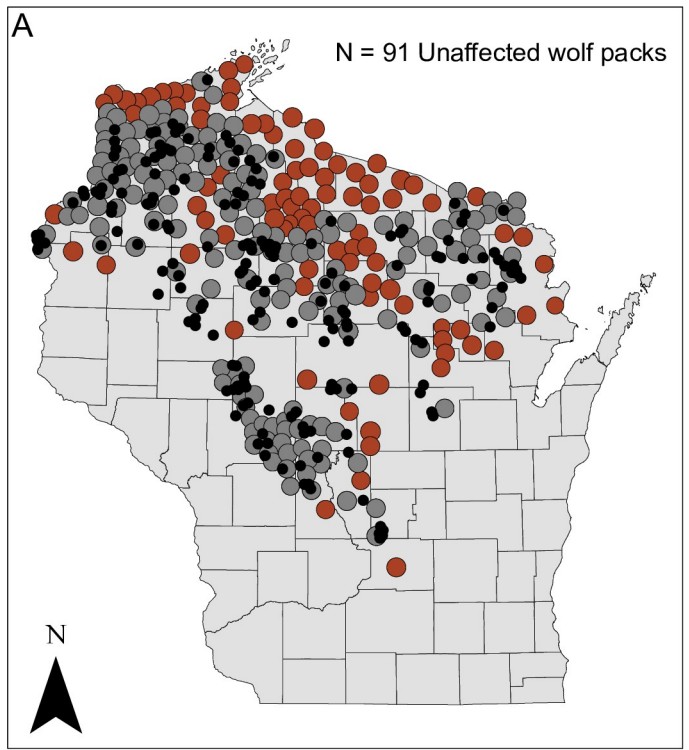
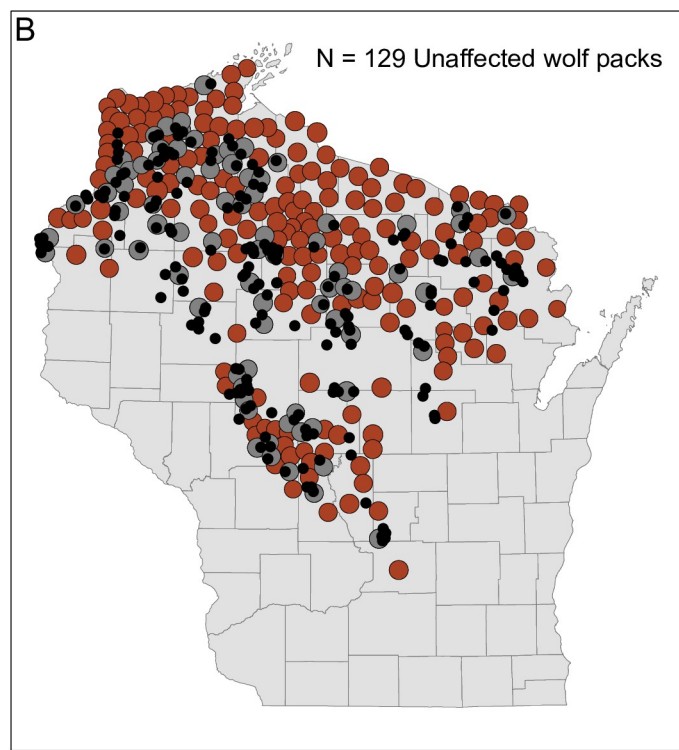

Fig 1. **Two scenarios for Wisconsin wolf packs affected by wolf-hunt.** (A) 91 breeding packs scenario: Any wolf kill location self-reported by hunters was extended by the average wolf territory size (161.3 km$^2$ according to [28]) and if it overlapped a wolf territory, those wolf packs were assumed not to have reproduced successfully. (B) 129 breeding packs scenario: Any hunter-reported wolf-kill location inside a wolf pack territory was assumed to have prevented that pack from reproducing successfully. To estimate the number of breeding wolf packs for these two scenarios, we used ArcGIS Desktop 10.7.1 to convert the map of 2020 Wisconsin wolf pack locations reported in [22] and the February 2021 self-reported wolf harvest location map from [27] into shapefiles. We then used spatial overlay and geo-rectification to find overlap in territories and self-reported kill locations. The Wisconsin county map was sourced from the WDNR Open Data Portal (https://data-wi-dnr.opendata.arcgis.com/).

provided incomplete and unclear data on deaths of wolves after 31 December 2011 [39–41,67–69] and particularly incomplete after 14 April 2012 [24,25,48,54,55,70,71]; S1 Fig 2.

To validate the estimate of D, we had separate published estimates using different methods for adult wolves from 1979–2012. For collared wolves only, the cumulative incidence of all endpoints (deaths or disappearances) for collared wolves 365 days after collaring was 0.42–0.52 depending on ESA listing status [39]. That study used time-to-event analyses in a competing risks framework. By contrast, a cruder estimate using a weighted average of collared and uncollared adult wolves dead as a proportion of the population size at the start of each wolf-year, which did not take into account time-to-event but considered uncollared wolves, estimated the rate at 0.18 for radio-collared wolves and 0.47 (SD 0.19 annually) for uncollared wolves [40]. Similarly, [72] reported higher mortality rates for uncollared Alaskan gray wolves. See also [73] for another large carnivore in which GPS collars are

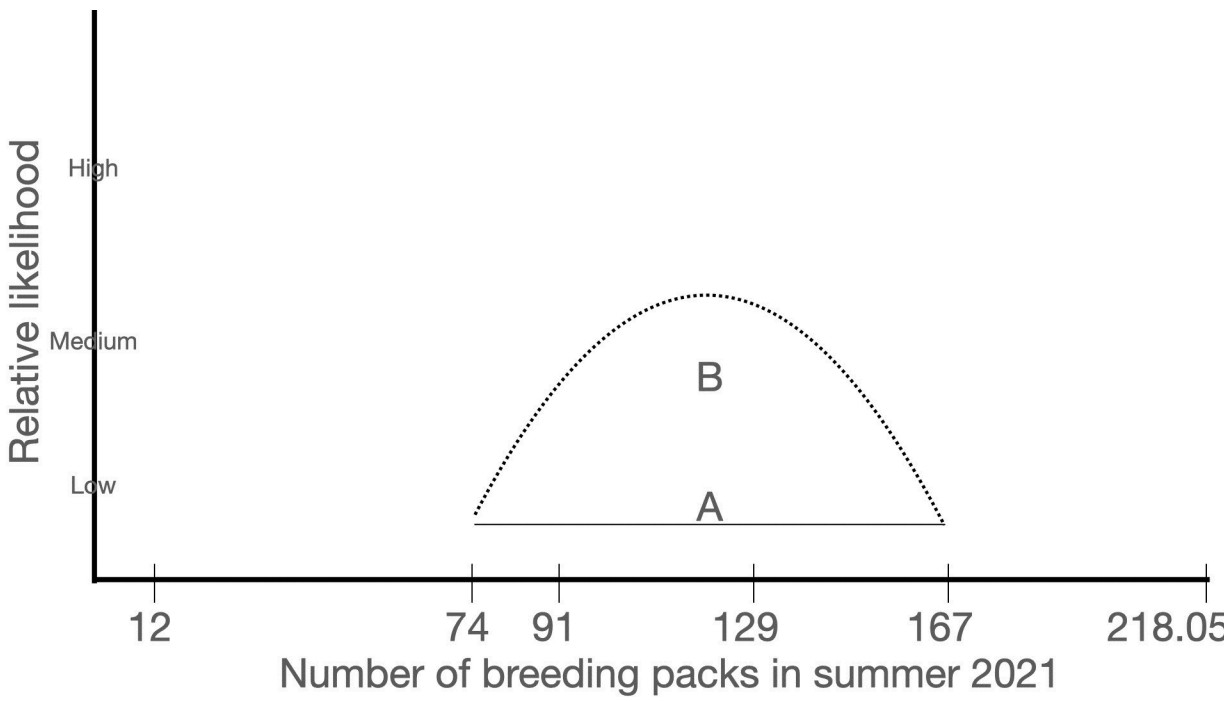

**Fig 2. Two ways to depict the uncertainty about the number of breeding packs.** We selected the uniform distribution (A) because we had no evidence to support the normal distribution (B). Also, the uniform, uninformative distribution allows the data to influence the result rather than our preconceived notions of what is typical in biological distributions. Similarly, we used a uniform distribution analogous to A to estimate deaths.

associated with higher survival. In 2020, approximately 5% of the wolf population was collared, so the weighted average annual mortality rate would be 0.46. The third peer-reviewed estimate of mortality covered the years 1979–2013 which included a wolf-hunt in Fall 2012. However that estimate it provided of 23.5% annual mortality for radio-collared adults in a time-to-event analysis [58] seems low. For instance, that study failed to account for several confounding variables and took unjustified steps in analyses. The unjustified steps were to include a variable for a change in slope in the year 2004 which is distinguishable only by the methods of analysis of census data [44,46]. And there were similar changes in census methods and methods of analysis in 1995, 2001–2003, and 2012, which [58] did not consider. We do not understand why 2004 was special and they did not explain why. Also, the authors lumped nonhuman causes of death with unknown causes of death, a step that several analyses have shown to be unjustified because time-to-event analyses show very different timing in the hazard of nonhuman and unknown causes [39–41,59]. Moreover, [58] did not acknowledge that uncollared wolves may have faced higher rates of mortality, or the multiple, corroborating lines of evidence showing that wolf survival and wolf population growth declined when ESA protections were lifted 7 times from 2003–2013 [12,39–41,51,74,75]. Finally, [58] did not account for the changes in incidence of wolf mortality with hound-trainign seasons, deer-hunting seasons, and bear-hunting seasons, especially elevated during months of snow cover [59]. Therefore, h [58] is certainly too low given the conditions between 3 November 2020 and 13 April 2021.

In sum, we had three published estimates of annual mortality rate from prior years ranging from 0.235–0.52 using three different methods on similar datasets, with which we could validate our estimate of D, at least qualitatively. We used a uniform distribution analogous to Fig 2 for D.

## Scenarios for wolf-hunt death tolls (H) and order of operations in our model

The last step in our analysis was to subtract H for the death toll from the uncertain wolf-hunt scheduled for November 2021. These death tolls assume zero sub-lethal injuries unreported as legal kills, and assuming zero additional cryptic poaching beyond that already captured in annual mortality rates during periods without ESA protections [23,39].

Uncertainty about the death toll reflects different permutations of the quota set by the DNR (130 wolves) and that quota voted by the NRB on 11 August 2021 (300 wolves) in addition to the following factors that might raise or lower the eventual death toll: over-kill in February 2021 of 99 or 82% might repeat itself; or the tribal treaty right to reserve 43% of the declared state quota (leaving a death toll of 74 if the DNR quota of 130 were to be implemented). Therefore, we modeled H as a continuous, normal distribution with a mean of 300 ranging from 0–600. H was our perfectly measured x variable on which to regress the population estimate using ordinary least squares algorithms. In Results and Discussion, we focus on three x values (0, 130, and 300) representing the preferred, legal death tolls for the plaintiffs [64,76], DNR, and NRB respectively. We also discuss a fourth death toll (74), which was the DNR's 130 death toll minus the tribal treaty right reserved 43%.

Because annual mortality rate is a proportion of living wolves, the order in which we deduct non-hunt deaths may be important. Subtracting the November wolf-hunt first would over-count deaths from other causes because these are calculated as a proportion using the annual mortality rate described above. However, half the year passes before the wolf-hunt and a smaller number of wolves (adults only) are present to die of such causes, so the number of deaths would be under-counted, if we deduct the non-hunt mortality first. Ideally, one would subtract the adult summer mortality, add pups surviving to November, subtract the wolf-hunt and then subtract adults and pups dying from other causes in the winter. However, we believe uncertainty about the other parameters described previously is far greater than the slight difference this more realistic algorithm would create. Therefore, to keep the calculations simple, we deducted all the annual mortality before the wolf-hunt, which treats the wolf-hunt as purely additive. The bias we introduce by estimating a higher number of non-hunt deaths is offset by the bias we have already introduced by dismissing unreported deaths and excess illegal killing. For example, the most rigorous study of cryptic poaching to date on the endangered Mexican wolf estimated that disappearances of collared wolves in this closely monitored population went up 121% when the wolf was not listed under the ESA, compared to periods of strict ESA protection [38]. However, we took the conservative step of not using this estimate or the higher mortality rate of collared wolves estimated in [39].

Finally, before evaluating legal thresholds, we subtracted 42 wolves living entirely or mostly on tribal reservations [27], because these are managed by the co-sovereign tribes whose governments declared wolves protected from public hunts [77].

Randomizing: Our modeling procedure used random generation of values for every parameter in Eqs 1 and 2 in 1200 iterations repeated once for each census method (traditional and new). We tripled that for the final estimates of $N_{2021}$ to 3600 iterations to boot-strap the distribution around the means. S1 Table provides the randomization outcomes and the distributions for each parameter. S2 Table provides the code.

## Results

Table 1 presents the estimate of annual rate of mortality, D, which ranged from 0.38–0.56 when we used the traditional census method or a range from 0.17–0.58, with the most likely values 0.38–0.48, when we used the new census method. Note these two methods have

**Table 1. Estimates of the annual mortality rate (D$_{2020}$) of Wisconsin wolves between 15 April 2020 and 14 April 2021.** We used two census methods to estimate N$_{2020}$ and N$_{2021}$ and reproductive parameter R (mean, lower and upper bounds of the 95% CI from [53] for 256 wolf packs. D is estimated as (N$_{2021}$-N$_{2020}$) divided by (0.5 * R$_{2020}$ + N$_{2020}$) following Eq 3. We assumed the mean value for N$_{2021}$ because the state did so for setting policy.

| Table 1. | Traditional census method (uniform distribution) estimating D$_{2020}$ | | New census method (unknown non-uniform distribution) estimating D$_{2020}$ | | |
|---|---|---|---|---|---|
| Estimates of D$_{2020}$*** | A* | B* | C** | D** | E** |
| Mean | 0.41 | 0.45 | 0.51 | 0.36 | 0.22 |
| Minimum bound | 0.38 | 0.43 | 0.50 | 0.34 | 0.17 |
| Maximum bound | 0.53 | 0.56 | 0.58 | 0.48 | 0.38 |

* For the traditional census method the minimum bound in 2020 (1034)—the maximum bound in 2021 (751+218) provides the values in column A and the maximum bound in 2020 (1057)—the minimum bound in 2021 (695+218) provides the values in column B.

** For the traditional census method, the state set policy used the mean in 2021 (1195–218), so we calculated variation by using the upper bound (1355) in column C, the mean (11995) in column D, and the lower bound (739) in column E.

*** The mean, minimum bound, and maximum bound reflect the mean and CI of R (see Methods).

different distributions. The former is uniform and the latter is unknown but extremely unlikely to be uniform. Given the new method has very wide bounds and hence great uncertainty and lacks peer reviewed validation as of writing, we have elected to view it qualitatively as consistent with the traditional method because its bounds entirely contain the bounds of the traditional method, Also, the latter is consistent with recent, peer-reviewed published estimates of annual mortality rates (see Methods). Therefore, in the next step we take D to be 0.38–0.56 with a uniform distribution.

## State wolf population N$_{2022}$

Figs 3 and 4 depict the probabilities of crossing legal thresholds for the Wisconsin wolf population. The slope of Fig 3A suggests that any death toll above 16 creates a better than average possibility of crossing the threshold of 350 wolves (state population goal). For the new census method (Fig 3B), that threshold is met at a death toll of 88 but the uncertainty is three times greater and the risk of crossing lower thresholds also increases. The probability of crossing the second threshold (state listing) exceeded 50% at death tolls of 113 and 189 wolves, for the traditional and new census methods respectively. The probability of crossing the thir threshold (state extirpation) exceeded 50% at death tolls of 359 and 443 wolves, for the traditional and new census methods respectively. The traditional census method had a reliable slope judged by its r-squared value, twice as reliable as the new census method (Fig 3A and 3B).

Even a death toll of zero might lead to the wolf population declining below the 1999 population goal of 350 (Fig 4). If the new census method were used, the distributions would be flattened raising the probability of undesirable thresholds.

The DNR asserted the tribal treaty right to 43% would be respected and the co-sovereign tribes that signed those treaties had asserted they would not hunt those wolves. Therefore, we examine the resulting death toll of 74 next. Using the traditional census method, N$_{2022}$ would average 329 (SD 44) wolves with a 1% probability of crossing the listing threshold of 251 and a 65% probability of crossing the state population goal of 350 (orange and yellow lines respectively in Figs 3 and 4). Using the broader, flatter distribution from the new census method, N$_{2022}$ would average 402 (SD 132) wolves with a 13% probability of crossing the listing

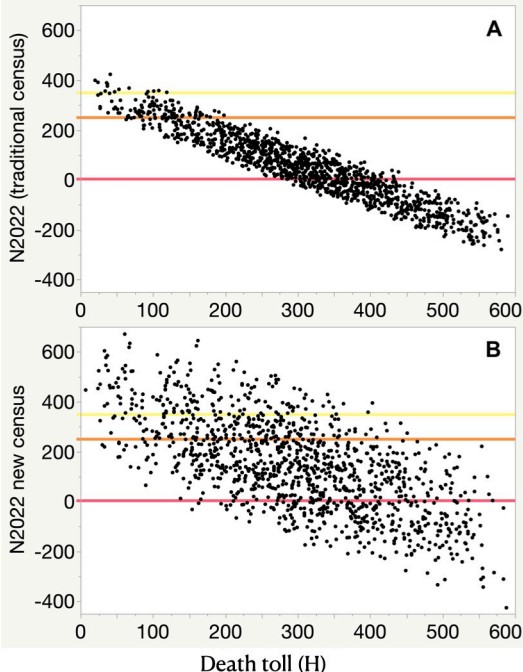

**Fig 3.** The relationship between wolf-hunt death tolls in Fall 2021 (x-axis) and predicted wolf population status in Wisconsin on 14 April 2022 (y axis). Ordinary least squares regression of $N_{2022}$ against H for the traditional census method (A, regression line not shown adjusted $r^2$ = 0.89, $N_{2022}$ = 366–1.016*H, SE slope = 0.010) and new census method (B, regression line not shown adjusted $r^2$ = 0.45, $N_{2022}$ = 437–0.983*H, slope SE = 0.032). We ran 3600 iterations for each panel, in which we randomly selected 1200 values for each parameter in Eqs 1 and 2. Three reference lines represent the legal thresholds of 1 (extirpation, red), 250 (state listing, orange), and 350 (state population goal, yellow).

threshold of 251 and a 36% probability of crossing the state population goal of 350 (orange and yellow lines respectively in Figs 3 and 4). The above averages and probabilities assume no over-kill or illegal kills beyond that estimated by our background mortality rate.

## Conclusions

We modeled a population of wolves recently removed from the USA list of endangered species, subjected to an unprecedented hunting season in February 2021, and proposed for another hunt in the winter of 2021–2022. We present this case, among other reasons, to illustrate the use of legal thresholds to define the probabilities that policy will result in undesirable effects. Societal value judgments have produced legal thresholds that decide what is precautionary and what is not, relieving scientists of the appearance of making personal value judgments when evaluating policy effects. We quantified the probabilities of crossing three legal thresholds with simple models and Bayesian concepts to account for uncertainty. We demonstrated constructive approaches to using a mix of qualitative and quantitative information to reduce uncertainty to manageable levels with uninformative, uniformly distributed prior information. The precautions we studied were set by legal thresholds so we could operationalize precautionary approaches without interposing our own values. For organisms at risk of extinction like in our case, precautions are relatively clear because hunting can only harm the targets, assumptions about resilience should be viewed as risky, and the sustainability of human actions should be viewed skeptically.

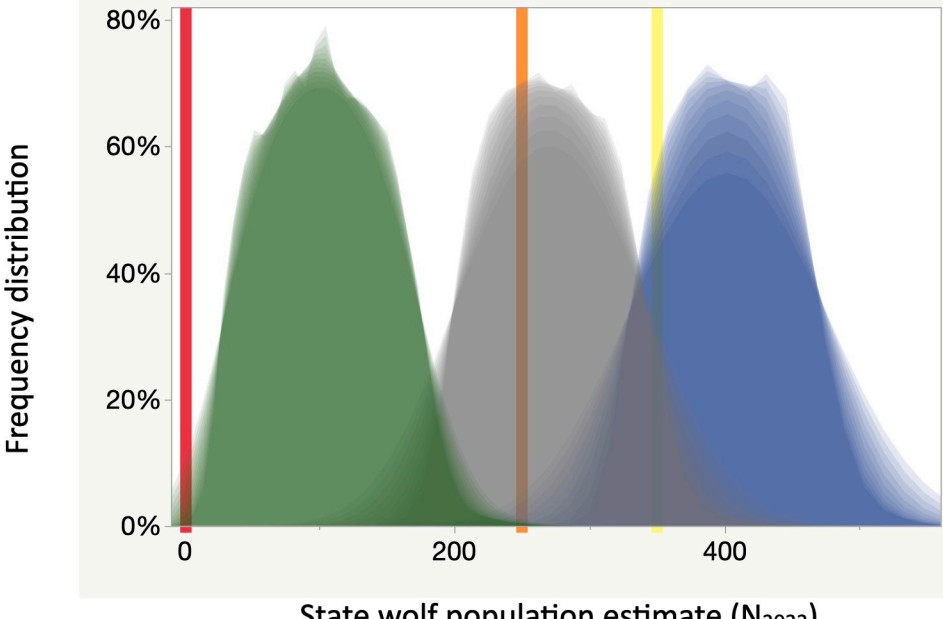

**Fig 4. Distributions of predicted population estimates for Wisconsin's wolves on 14 April 2022.** Frequency distributions assume death tolls of 300 (green), 130 (gray), and 0 (blue) relative to reference lines of extirpation (red), listing (orange), and population goal (yellow). We ran 3600 iterations to generate smoother probability distributions as "shadow grams" made in JMP® 15.0, 2021, for each value of H. These distributions rely on the traditional census method (Fig 3A) and average and SD follow: (green) 61 SD 44 with a 9% chance of extirpation and 100% chance of dropping below the state listing threshold, (gray) 231 SD 45 with a >99.5% chance of dropping below the state population goal and a 64% chance of dropping below the state listing threshold, (blue) 361 SFD 44 with a 13% chance of falling below the state population goal.

Several new results emerged for Wisconsin's wolves. We report high probabilities that a second wolf-hunt in winter 2021–2022 would drive the Wisconsin wolf population to undesirably low levels, judged by legal thresholds and the current quotas recommended or set by the state. Moreover, a repetition of the over-kill of the February 2021 wolf-hunt (by 99 wolves or 182% of the legal quota) risks extirpation of the state population leaving only wolves in tribal reservations. Even a well-regulated wolf-hunt at the quota level recommended by the state wildlife agency (130) is more likely than not to require statutory listing on the state endangered and threatened species list. We found any wolf-hunt in November 2021 poses a measurable risk of an undesirable outcome and any quota >16 wolves is more likely than not to lead to an April 2022 wolf population below the threshold of the 1999 population goal [43]. Therefore, no wolf-hunt is safe when viewed from a precautionary viewpoint. We also present the first estimates for annual mortality rate between 15 April 2020 and 14 April 2021. That rate per year was 0.38–0.56 adults and young of the year that survived to November. If we add the February 2021 wolf-hunt to the latter rate, the total annual mortality rate in 2021 would rise by >0.18 (218 / 1195). The sum of those two rates seems unsustainable, even if we accept a nonhuman-caused rate of mortality of 0.09 [45]. The resulting one-year mortality rate of 0.56–0.74 in Table 1 is too high to be sustainable by any of the credible estimates in the literature reviewed by [31]. Also, Table 1 annual mortality rates are substantially higher than the DNR "consensus" estimate of 13% [23] plus approximately 9% nonhuman-caused. Therefore, we reject the DNR's consensus method for estimating mortality as unscientific and highly inaccurate. Furthermore, the range of annual mortality rates in Table 1 was almost never so low as estimated by [45]. Their estimate of 23.5% is only plausible for 2020 if one accepts a drastic rise in

population size from 2020 to 2021, which no authority has claimed. As predicted by [36], the February 2021 wolf-hunt seems to have led to an increase in wolf-killing in response to alleged predation on domestic animals. Also as predicted by [36], reducing protections for wolves increases calls for legal killing; see also [46]. Reducing protections leads to lower survival for wolves when all causes of death are considered [36]. Therefore, we recommend the state halt lethal management of wolves in years it plans wolf-hunts because we see no method or regulation in place to deduct state lethal control totals from legal quotas. We also recommend the state revise its estimate of mortality and in so doing also publish all mortality data in a scientific manner including distinguishing between radio-collared wolves and others with time on the air for the former. For all governments reporting wolf mortality, we recommend more care in estimating poaching and the use of forecasting methods that take into account a spike in legal mortality after governments lower protections for imperiled species [38]. Also we recommend wolf managers focus on poaching enforcement when seasons for hunting other (non-wolf) large mammals are open [59]. These recommendations probably apply as well to other controversial wildlife.

## Bridging science and policy when both are controversial

Our topic is controversial in wildlife management science and in public policy. Below we discuss how values in wolf policy affect the handling of precautions and how controversies in science affect handling of uncertainty. The foundations of the controversies are diverse values toward wolves in the USA [78,79], mirrored elsewhere [20,80]. These publics do not simply diverge quantitatively in their support of wolves but qualitatively, differing in mutualism values that favor non-lethal coexistence [81]. Naturally, such public debates affect government agencies charged with managing wildlife.

In the USA, wildlife agencies are typically allied to hunters [82,83]. Regardless of its origins, the status quo in all but a few states (California and Colorado currently) that host gray wolves is towards liberalizing wolf-killing. States such as Wisconsin repeatedly moved towards public, regulated hunting, trapping, and hounding for the past 23 years [46]. Those values embraced by the agency push against the above-mentioned shift in public values. State wildlife policies also clash with scientific evaluations.

Several governments' legal wolf-killing quotas exceed levels deemed sustainable by scientists who cite the agencies for non-transparent handling of uncertainty or data [23,33,37,46]. High quotas for killing large carnivores such as wolves, bears, big cats, have sometimes been associated with undue political pressures on the agencies. One manifestation of such political pressures is the tendency for agencies to report unrealistic biological parameters that appear to the uninformed to support claims that killing is 'sustainable' or 'safe'. Such "political populations" [30] seem designed to satisfy political demands by inflating population parameters of the carnivores targeted for killing. A recent review of 666 North American wildlife hunting plans found a large majority of the plans lacked hallmarks of scientific process such as setting clear objectives, independent review, and transparency about data or methods [84,85]. Regrettably, the Wisconsin wildlife agency got high marks for past management in the latter review. Our work suggests those high marks were not merited then or now [23,46]. We report here that the state of Wisconsin created a political population, by the above definition, when it set quotas for a second wolf-hunt in one year without data on reproduction or poaching in the 11 months prior. Such inflation or other distortions of sound science-informed management seem to surface when agencies are not required by law to use best available science defined by third parties, but rather can pick and choose the evidence they wish to use based on their personal or organizational values [10,12,46,86–88].

The politics that led to the current situation in Wisconsin are complex and go beyond a pro-wolf and anti-wolf dichotomy. In brief, a state wildlife agency (DNR) under the executive branch led by the governor appears to be clashing with the commission (NRB) whose members are appointed by governors but confirmed by the legislative branch. Those two bodies clashed publicly over wolf policy in August 2021 (https://www.wpr.org/listen/1836191, accessed 17 August 2021;[25]). Besides that intra-governmental clash there is a long-standing intergovernmental dispute between the state and the co-sovereign tribes of the region who have federal treaty rights to half of almost all natural resource extraction. The state and tribes have co-managed a subset of resources relatively amicably under federal treaties, but walleye fish and wolves have been a point of friction for over a decade [77,89,90]. The Red Cliff tribal government and other tribal governments that signed those treaties filed a federal lawsuit on 19 September 2021 alleging treaty rights violations during 2021 wolf-hunt rule-making [64]. Besides being pro-wolf, tribes in our region are also pro-hunting for subsistence, spiritual, and traditional uses, which represents a distinct set of values in the broader public. Consistent with the controversial nature of our topic, the Wisconsin wolf-hunt under consideration here is the subject of lawsuits instate court [76] and federal court [64].

The state case led to a temporary injunction barring the sale of permits to hunt wolves based on the judge's decision that the state wildlife agency acted unconstitutionally [91]. Although legal decisions generally reflect only a court's interpretation of the law, the ongoing state court case also raises issues of science that concern us here. The state court agreed with plaintiffs on the need to delay the case [92], when the plaintiffs brought to the court's attention that the state had filed an incomplete administrative record [93]. A complete record of all comments and other materials submitted to the agency by the public is required by law, following the Wisconsin Supreme Court decision of Lake Beulah Management District. The Supreme Court advised the public to "submit evidence to the agency decision makers while they are deciding what action to take" p.7, [94], so that they can "ensure that information will be considered by an agency in its decision making and will be included in the record on review. . ." p.355, [94]. The plaintiffs identified 59 instances where comments from scientists and the public were missing from the administrative record under review by the state court [93]. The plaintiffs' implied that the administrative record was preferentially full of gaps that had been submitted by scientists and scholars critical of the proposed wolf-hunt (p.5 [93]. In sum, the state wildlife agency in this case has in part created a political population of wolves by ignoring contradictory scientific evidence and commentary. In our context, the above elements of controversy about Wisconsin's wolves underline another point about uncertainty and precaution.

When public comments opposing killing policies or otherwise encouraging caution are dismissed or omitted from the administrative record, the government creates an illusion that its plans are supported by the public and an illusion that is plans are cautious, because dissenting voices were silenced. Furthermore, dismissal or omission of scientific evidence that undermines the government's assertions of fact seem to treat scientific uncertainty as something that can be willed away through political might. Scientists should speak out against scuch handling of scientific information by governments. The above-referenced controversies among publics, within the scientific management community, and between managers and decision-makers highlight that neither science of uncertainty nor values towards precautionary approaches alone are at play.

## Recommendations for scientific management

We recommend scientists account transparently for uncertainty so that decision-makers can apply precautionary approaches to public policy. Scientific uncertainty often hinders

precautionary approaches. Yet policymakers are often forced to decide anyway. If scientists turn away from public policy debates characterized by wide gaps in data or great uncertainty, then decision-makers may decide based on opinion, anecdote, or political pressures. We aimed to bolster scientists' confidence in their ability to grapple with uncertainty in a way useful to public policy. We recommend that scientists practice analysis and communication that improves their ability to explain what the uncertainty means for policy and the public.

A common thread running through our work is that the more uninformative the prior data, the more scenarios one should present and the more transparent the assumptions about inputs should be. This recommendation aligns with our inclination to use a simple model so that non-specialist members of the public and decision-makers can easily explore and adjust inputs. Any reader can follow our lead and estimate the outcomes for any death toll they prefer. Also, we avoided the critique of precautionary approaches articulated by Curtis (see Introduction) by sticking to peer-reviewed evidence wherever available, evaluating that evidence transparently, and when unavailable we used uninformative, uniform distributions on priors to account for gaps in important data. Our results speak to how precaution can be operationalized even with high uncertainty about data.

The 82% over-kill seen in <3 days during the February 2021 wolf-hunt has raised national debate about the security of state wolf populations. That hunt and our calculations here suggest hunters and poachers can extirpate a relatively small wolf population, in short order and without poison, which contradicts an unsubstantiated assumption that poison would be needed to eradicate wolf populations [95]. We expect proponents of that assumption will claim that the Wisconsin wolf population would persist in tribal reservations, that it would be rescued by neighboring states, or claim that we were too pessimistic. However, such arguments miss the point. Anyone who steps away from the precautionary approach must present stronger evidence for their more optimistic view. The uncertainty grows when one takes optimistic views because the more extreme higher values produce greater intervals between minimum and maximum bounds (because we were bounded by zero in this small population of wolves). Therefore, the burden of proof and demands for data is heavier for those who advocate for killing.

## Supporting information

**S1 Fig. Unpublished figures by WDNR staff on 8 April 2021 during a public presentation to the Wolf Harvest Planning Committee.** Fig 1 shows unpublished results of the new census method. Fig 2 shows how mortality data for April 2020-April 2021 were presented. (PDF)

**S1 Table. Outputs of randomization for each variable in Eqs 1–3.** Table headers describe the distributions used in randomization. Yellow cells are not user-defined but rather outputs of randomization or outputs of equations. White cells are user-defined so a user can enter different death tolls (H). These are provided for the purposes of exact replication of our results. See S2 Table for algorithms. Yellow fields are generated randomly, white fields are input by the user, and gray fields are outputs of algorithms associated with the white input fields. The attached example is of H = 74 input by the user. (PDF)

**S2 Table. Algorithms used in randomization and modeling scenarios, showing formulae in Apple Numbers⑬ 2021 v11.2.** Note that the formulae should follow the insertion of ' = ' to become active and then should be pasted into all cells within a sheet. Yellow sheets contain outputs of randomization, whereas white or gray sheets are user input fields. S2 Table presents the outputs of these algorithms in a single iteration used in the Results (1200 values) whereas

results in Fig 4 represent three such iterations (3600 values).
(PDF)

## Acknowledgments

We thank F. J. Santiago-Ávila for comments on the revision.

## Author Contributions

**Conceptualization:** Adrian Treves.

**Data curation:** Adrian Treves.

**Formal analysis:** Adrian Treves.

**Funding acquisition:** Adrian Treves, Naomi X. Louchouarn.

**Investigation:** Adrian Treves.

**Methodology:** Adrian Treves.

**Project administration:** Adrian Treves.

**Resources:** Adrian Treves.

**Validation:** Adrian Treves.

**Visualization:** Adrian Treves.

**Writing – original draft:** Adrian Treves, Naomi X. Louchouarn.

**Writing – review & editing:** Adrian Treves, Naomi X. Louchouarn.

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
