## [Decision Letter · Decision Letter 0]

26 Jan 2022

PONE-D-21-32862

Uncertainty and precaution in hunting wolves twice in a year

PLOS ONE

Dear Dr. Treves,

Thank you for submitting your manuscript to PLOS ONE. After careful consideration, we feel that it has merit but does not fully meet PLOS ONE’s publication criteria as it currently stands. Therefore, we invite you to submit a revised version of the manuscript that addresses the points raised during the review process.

The reviewers raised only a few questions regarding this manuscript. They suggest a revision of figures and several supplementary explanations or rewording in the methodology. I also believe that the manuscript is of good quality; however, some parts require improvement. For example, please revise the references, as there are many errors (Tva v hill,  Epstein Y. Killing wolves to save them? ,  Johnson RR, Curtis A...).  Furthermore, the Wikipedia article should be archived (https://archive.org/web/ and referenced with the link to the archived article. The introduction is followed by a chapter named Uncertainty. It is the intention, or should it be part of the intro? You stated in the letter to Plos One that the work will be controversial. I would like to introduce a short discussion on what is controversial in the paper and what you expect to trigger with this paper. This info is in the paper, but I think it deserves a more straightforth acknowledgment.

We look forward to receiving your revised manuscript.

Kind regards,

Laurentiu Rozylowicz, Ph.D.

Academic Editor

PLOS ONE

***Request from the Editorial Staff:

PLOS ONE requires that all conclusions in the manuscript must be supported by presented data (https://journals.plos.org/plosone/s/criteria-for-publication#loc-4). In this instance, please amend any statements, such as the last couple of sentences in the abstract, to ensure that they are not opinion based. In addition, please ensure that the conclusions are based on the evidence shown here, staying away from political discussions and/or recommendations as much as possible.

Journal Requirements:

The authors declare no financial competing interests.

AT discloses the following non-financial, potential competing interests.

Professional service to organizations or editorial boards 

Board of director (unpaid):

President, Future Wildlife (2020), Board member Wildlife for All (Sep. 2021-present) 

Science advisor (unpaid): 

Project Coyote (2012–)

Northeast Wolf Coalition (2014–)

Endangered Species Coalition (2016–)

Friends of the Wisconsin Wolf (2015–)

Living with Wolves (2016–)

Rocky Mountain Wolf Coalition (2018–2021)

Earth and Animal Advocates (2019–)

Benton County’s Agriculture and Wildlife Protection Program (2018–)

Wild Earth Guardians (2020–)

Member (unpaid):

Union of Concerned Scientists (2015–),

IUCN Bear Specialist Group task-force on human-bear conflicts (2012), IUCN Wolf specialist (2016–), 

Public Employees for Environmental Responsibility (2015–2019).

Expert declarations (unpaid):

Wi Federated Humane Societies et al. v Stepp. 2013. WI Court of Appeals District IV; 

WEG v Colorado Parks and Wildlife Commission et al. 2017. District Court, Denver Country, Colorado; 

Western Watersheds Project et al. v USDA Wildlife Services. 2018. U.S. District Court for the District of Idaho 1:17-cv-00206-BLW Doc 22-3; 

CBD & Cascadia Wildlands v WDFW 2018. Superior Court of Washington for Thurston County. 18-2-04130-34. 

CBD v WDFW et al. 2019. Superior Court of Washington for Thurston County, 18-2-02766-34. 

Huskin et al. v WDFW et al. 2019.

 Superior Court of Washington for King County 19-2-20227-1 SEA.

Great Lakes Wildlife Alliance et al. v. Cole et al. Circuit Court Dane County, Wi 2021cv002103 Branch 9 Circuit Court Summons Dane County, WI, Case 2021CV002103 Document 5 Filed 08-31-2021

Paid service as external reviewer or speaker: 

Swiss-NSF SPARK (2019), Antioch University of New England (2018–2020), Landmark Foundation (2017), various publishers (2007–2017), U.S. Fish & Wildlife Service (2019), French Ministry of Environment, Scientific Council on Wolves (208-present), Ministry of Environment, Alfred Toepfer Academy for Nature Conservation, Lower Saxony, Germany (2021-present), NABU, Germany (2015, 2021)

4. Please amend the manuscript submission data (via Edit Submission) to include author Louchouarn, Naomi X.

5. We note that you have referenced (ie. Bewick et al. [5]) which has currently not yet been accepted for publication. Please remove this from your References and amend this to state in the body of your manuscript: (ie “Bewick et al. [Unpublished]”) as detailed online in our guide for authors

6. We note that Figure 1 in your submission contain [map/satellite] images which may be copyrighted. All PLOS content is published under the Creative Commons Attribution License (CC BY 4.0), which means that the manuscript, images, and Supporting Information files will be freely available online, and any third party is permitted to access, download, copy, distribute, and use these materials in any way, even commercially, with proper attribution. For these reasons, we cannot publish previously copyrighted maps or satellite images created using proprietary data, such as Google software (Google Maps, Street View, and Earth). For more information, see our copyright guidelines: http://journals.plos.org/plosone/s/licenses-and-copyright.

Reviewers' comments:

Reviewer's Responses to Questions

**Comments to the Author**

1. Is the manuscript technically sound, and do the data support the conclusions?

Reviewer #1: Yes

Reviewer #2: Yes

2. Has the statistical analysis been performed appropriately and rigorously? 

Reviewer #1: Yes

Reviewer #2: Yes

3. Have the authors made all data underlying the findings in their manuscript fully available?

Reviewer #1: Yes

Reviewer #2: Yes

4. Is the manuscript presented in an intelligible fashion and written in standard English?

Reviewer #1: Yes

Reviewer #2: Yes

5. Review Comments to the Author

Reviewer #1: General comments on the manuscript

It was a pleasure reading this manuscript. The manuscript is constructed to allow science to have a practical input therefore the conclusions are supported by the data, while ethical and logical arguments are supporting the study objectives and the discussion. The fact that the manuscript is actually describing the uncertainties in relation to precautions to explain how one can get from basic data to conclusions plays a methodological and an informal role. Since one of the manuscript objective is to inform not only scientist but also wildlife managers and policy makers on social sensitive topics like hunting quotas keeping the statistical tools at a basic level is highly relevant for the manuscript and it is a good editorial strategy.

General suggestions for the authors

I believe the title is not reflecting at all the second goal of the manuscript related to the potential of the method being replicated for other species. I would suggest a new title to incorporate the recommendations for scientist maybe something like: "Uncertainty and precaution in hunting management should build a bridge between science, politics and decision making."

I believe your work is a perfect example to explain how the concept of "political population" (Darimont et al. 2018) works in practice and also how cultural differences lead to different perspective on the same object, in this case how the Red cliff government has a different perspective. I would recommend to explore more this two ideas in the Introduction and also in the Discussion. In the end Uncertainty is also a result of social and professional culture. From my understanding the concept of political population is a powerful concept and used wisely can support strategically the influence of science on policies. On relation to this proposal I believe it will be more clear if you would explain the three thresholds (<2, <251, <350) separately and not in the Uncertainty section in the Introduction since they are social and political justified.

I would suggest also to find a way to explain graphically all the values and ranges for your estimation of potential values for R an M to help the readers to follow the algorithm easily.

Specific comments on the manuscript

line 72 - Without speculating I would mention main causes of dead for 98-105 wolves.

line 238 - why did you assume only 218 wolves and not use also the other 99 overshot? please explain.

line 295 - the value of harvested wolves is 218 and the number of packs that could have breed is also 218 and this creates a little bit of confusion in the manuscript. Please find a way to eliminate this confusion.

296 - not sure if the 218 packs failed to bread or were potential breeding. In the first scenario and compared with the line 292 statement and following results seems that the second context is the right one. Can you please check this statement?

line 503 - please check the number of the figure.

line626 - maybe it is driven

Conclusion

I believe the manuscript is suitable for publishing in PLOSone, fulfilling the journal criteria and since it covers methodological aspects that can be replicated and it is promoting the improvement of research quality it is highly relevant for several professional groups. Really hope to see this beautiful mixed of math science, ethics and anthropology published and used by scientist to promote better policies.

Reviewer #2: The article tackles a highly important and critical aspect of wolf (and wildlife in general) harvesting decision-making processes. Hunting animals with a relatively slow life history and a relatively high ecological function (such as large carnivores, which could trigger ecosystem-level changes) necessitates caution and thoughtful consideration. Because these processes are sometimes driven by political interests rather than scientific evidence, a critical and robust assessment of the effects of harvesting is vital for species persistence. In this context, the article makes an interesting contribution by providing a qualitative and quantitative analysis of the effect of harvesting Wisconsin wolves using both outdated quotas and adding an additional hunting season.

Despite the lack of reliable demographic data, the authors put up a significant amount of work in parameterizing the model. The authors performed a remarkable effort in finding the best possible parameters from different sources (i.e. grey literature) and compiling them in a meaningful way. Both immigration and emigration data, as well as non-linear relationship shapes, are not taken into account (yet critical in modelling population trends), but the authors present a good explanation for this methodological decision. The use of an uninformative uniform distribution helped reduce potential biases and, importantly, the authors used bootstrapping to account for parameter uncertainty, which is essential for the final interpretation of the results.

The manuscript is overall well organized and easy to read, it is well structured logically and linguistically, while the arguments are supported by appropriate literature. There are no major comments on the methodology used as is fully in line with the scope of the manuscript. As indicated by the authors, the work will most certainly be divisive due to its nature and message (i.e. zero death toll). While the statements are likely to be interpreted as political (particularly “Politics of precautions and the role of science'') , I believe the authors gave a reasonable justification for their standpoint.

The following comments are offered with the purpose of improving the manuscript:

Figure 1 - Add scale bar and north reference. Add A) and B) for consistency with the figure caption. The use of grey for both “affected wolf pack” and administrative borders (county?) is misleading. Either refer to the grey circle in the legend for “affected wolf pack”, or change color. Could the author report the average wolf territory size used for the spatial overlay and geo-rectification?

Figure 2 - The figure caption seems to be missing. The space interval between values in the x-axis is completely arbitrary, and thus incorrect. Regardless of the figure's theoretical basis, manipulating the intervals artificially is to be avoided.

Figure 4 - y-axis values are missing.

6. PLOS authors have the option to publish the peer review history of their article (what does this mean?). If published, this will include your full peer review and any attached files.

Reviewer #1: No

Reviewer #2: No

---

## [Author Response · Author response to Decision Letter 0]

30 Jan 2022

we uploaded a document with detailed responses.

---

## [Decision Letter · Decision Letter 1]

14 Feb 2022

Uncertainty and precaution in hunting wolves twice in a year

PONE-D-21-32862R1

Dear Dr. Treves,

We’re pleased to inform you that your manuscript has been judged scientifically suitable for publication and will be formally accepted for publication once it meets all outstanding technical requirements.

Kind regards,

Laurentiu Rozylowicz, Ph.D.

Academic Editor

PLOS ONE

Additional Editor Comments (optional):

Reviewers' comments:

Reviewer's Responses to Questions

**Comments to the Author**

1. If the authors have adequately addressed your comments raised in a previous round of review and you feel that this manuscript is now acceptable for publication, you may indicate that here to bypass the “Comments to the Author” section, enter your conflict of interest statement in the “Confidential to Editor” section, and submit your "Accept" recommendation.

Reviewer #1: All comments have been addressed

Reviewer #2: All comments have been addressed

2. Is the manuscript technically sound, and do the data support the conclusions?

Reviewer #1: Yes

Reviewer #2: Yes

3. Has the statistical analysis been performed appropriately and rigorously? 

Reviewer #1: Yes

Reviewer #2: Yes

4. Have the authors made all data underlying the findings in their manuscript fully available?

Reviewer #1: Yes

Reviewer #2: Yes

5. Is the manuscript presented in an intelligible fashion and written in standard English?

Reviewer #1: Yes

Reviewer #2: Yes

6. Review Comments to the Author

Reviewer #1: Dear Authors,

I acknowledge the changes made to your manuscript and I believe this second manuscript is more clear and your work is better presented in order to inform your target audience. My recommendation to the Editor is to agree with the publishing of your work mainly the following reasons (1) it is informal for wildlife managers and researchers, promoting a simple approach that can be replicated on your study area or on other part of the world where reliable demographic data is missing, (2) promotes the role of scientific input on wildlife management and (3) scientist should not avoid joining into debates on controversial topics.

Nevertheless, I believe your work might be contested by policy makers as being partisan, and in this scenario the paragraph Recommendation for scientific managements is essential. I would suggest, off course if possible at this stage of the manuscript, to advocate in one sentence for the acceptance by other parties/stakeholders of the scientific community as a player and not as spectator, judge or reviewer since: (1) You already mentioned BAS in the Introduction as a ”political” requirement and it make sense to come back to this statement and (2) Second science should have the potential to support deep leverages for wildlife management (Hartel et al 2019, https://doi.org/10.1111/cobi.13334).

Wish you to have a lot of citation (pro and against) with this paper!

Reviewer #2: I don't have any further comments. The authors have addressed all the comments and the article has improved as a result.

7. PLOS authors have the option to publish the peer review history of their article (what does this mean?). If published, this will include your full peer review and any attached files.

Reviewer #1: No

Reviewer #2: **Yes: **Andrea Corradini

---

## [Editor Report · Acceptance letter]

21 Feb 2022

PONE-D-21-32862R1 

Uncertainty and precaution in hunting wolves twice in a year 

Dear Dr. Treves:

I'm pleased to inform you that your manuscript has been deemed suitable for publication in PLOS ONE. Congratulations! Your manuscript is now with our production department. 

Kind regards, 

on behalf of

Dr. Laurentiu Rozylowicz 

Academic Editor

PLOS ONE